# Hot Compression Behavior of New Al-6Mg and Al-8Mg Alloy with Improved Hot Workability Fabricated by Direct Chill Casting Method

**Nam-Seok Kim [1,2]**, **Kweon-Hoon Choi [1]**, **Seung-Yoon Yang [1]**, **Seong-Ho Ha [1]**, **Young-Ok Yoon [1]**, **Bong-Hwan Kim [1]**, **Hyun-Kyu Lim [1]**, **Shae K. Kim [1,\*]** and **Soong-Keun Hyun [2,\*]**

[1]  Korea Institute of Industrial Technology, Incheon 21999, Korea; kimns@kitech.re.kr (N.-S.K.);
     kchoi74@kitech.re.kr (K.-H.C.); sy8357@kitech.re.kr (S.-Y.Y.); shha@kitech.re.kr (S.-H.H.);
     veryoon@kitech.re.kr (Y.-O.Y.); bonghk75@kitech.re.kr (B.-H.K.); hklim@kitech.re.kr (H.-K.L.)
[2]  Department of Materials Science and Engineering, Inha University, Incheon 22212, Korea
\*   Correspondence: shae@kitech.re.kr (S.K.K.); skhyun@inha.ac.kr (S.-K.H.); Tel.: +82-32-850-0412 (S.-K.H.)

**Abstract:** A hot compression test of new Al-6Mg and Al-8Mg alloys was conducted to understand the dynamic recrystallization (DRX) behavior by Mg contents. To investigate the hot workability of Al-Mg with high Mg contents, the hot deformation behavior of Al-6Mg and Al-8Mg alloys was analyzed by a hot compression test in the temperature range of 300–450 °C, and the strain rate range of $10^{-3}$–$10^{0}$/s. Subsequently, high-temperature deformation behavior was investigated through the processing map and microstructure observation. In this study, the results have shown that, as the Mg contents increase, the maximum and yield strength increase while rapid flow softening after the peak strain has been observed due to accelerated dynamic recrystallization (DRX). Finally, the increase of Mg contents affects an increase of heat dissipation efficiency to be an indicator of regular deformation.

**Keywords:** Al-Mg alloys; flow curve; hot compression test; processing map; microstructure; dynamic recrystallization (DRX)

## 1. Introduction

Since the last century, many researchers have investigated a method to improve the fuel efficiency of the automotive industry. In order to improve the fuel efficiency, the weight reduction of vehicles has been considered as one of the methods. Al alloys are considered good substitutes for conventional materials, such as steel, due to its much lighter weight. Specifically, 5xxx series (Al-Mg) alloys have decent properties for structural materials [1,2]. For example, Al-Mg alloys have high strength, corrosion resistance, ductility, and lightweight. As the early stage of weight reduction research for the vehicle, cast parts such as engine blocks in automobiles were replaced first. Gradually, the substitution has been expanded to other parts such as bumpers, hoods, fenders, roofs, and doors [3,4]. However, Al-Mg alloys typically exhibit relatively poor formability than steel because of their plastic anisotropy, high stacking fault energy, and negative strain rate sensitivity (nSRS), which causes instability during low temperature deformation [5–7]. High Mg content in Al alloys is not easily fabricated by conventional manufacturing processes because of their accelerated oxidation rate and high deformation resistance at elevated temperatures. Furthermore, in general, Mg solutes in Al-Mg alloys are distributed not uniformly, so that the strain anisotropy occurs [8,9]. In order to overcome these difficulties, several Al-Mg with high Mg hot working technics have been investigated in past years [10–12]. Nevertheless, the optimum hot working conditions of Al-6Mg and Al-8Mg have not been investigated and discussed sufficiently.

Hot deformation behavior of metals is determined by several parameters, such as temperature, strain rate, deformation mode, and flow stress [13–15]. In addition, as suggested

by Prasad et al. [16], the heat dissipation efficiency is the ratio of the energy consumed for material deformation over the total input energy during the deformation. Comprehensively, Prasad has proposed that the heat dissipation efficiency does not significantly change with the strain. However, as the amount of thermal restoration, like dynamic recovery and dynamic recrystallization, varies with the strain level. The flow behavior can be changed during hot working processes. The Mg content effect also needs to be considered to interpret the hot working properties of Al-Mg alloys. Therefore, the two different Al-6Mg and Al-8Mg alloys are investigated in this study.

In particular, the hot deformation behavior of the two Al-Mg alloys is estimated by analyzing flow curves, heat dissipation efficiency, and microstructures. The processing map derived by flow stress-strain curves is used to obtain the heat dissipation efficiency. Then, in order to interpret the flow data and processing map, the microstructure of each deformed alloy is observed by an optical microscope (OM) and electrical backscattered diffraction (EBSD).

## 2. Materials and Methods

### 2.1. Manufacturing Al Alloys for Specimens

In this study, the Al-Mg binary alloys (6, 8 wt% Mg) were cast as 7-inch billets by the direct chill casting method. In this casting method, Mg-Al$_2$Ca ingots, which show outstanding oxidation resistance when replacing conventional Mg ingots to fabricate Al-Mg alloys, were used, and these alloys were cast in a 5-ton scale using mass production equipment. The chemical composition of all alloys investigated in this study was analyzed by a spark optical emission spectroscopy (SPECTROMAXx, Spectro, Ameteck, Berwyn, PA, USA) and the chemical data are given in Table 1. The as-cast billets were homogenized by a two-step heat treatment: first 420 °C for 8 h, and then 490 °C for 8 h.

**Table 1.** Chemical composition (wt%) of Al-6Mg and Al-8Mg.

| Alloy | Mg | Ti | Ca | Al |
|-------|------|------|------|------|
| Al-6Mg | 6.22 | 0.03 | 0.06 | Bal. |
| Al-8Mg | 7.82 | 0.03 | 0.08 | Bal. |

### 2.2. Compressive Test

Compressive tests of Al-6Mg and Al-8Mg alloys were conducted to estimate their hot workabilities. The specimens for compressive tests were 15 mm in height and 10 mm in gauge radius. The Gleeble 3500 (Dynamic Systems Inc., Poestenkill, NY, USA) was utilized for this compressive test. The hot compressive tests were performed at several high temperatures (300, 350, 400, and 450 °C) and various strain rates ($10^{-3}$, $10^{-2}$, $10^{-1}$, and $10^0$/s). Strain was used by calculating the engineering strain obtained by measuring between anvils as the true strain. Flow curves in this study were given as the plot between true stress and true strain. Compression tests were finished when the total engineering strain reaches 0.6 (the same as 0.9 in the true strain). The heating rate was set to 10 °C/s. Then, the furnace temperature was maintained for 180 s to obtain the temperature homogeneity over the whole sample and the furnace. Additionally, the reproducibility of data was checked by repeating the same experiments at least three times. The original hot compression data were modified by compensating adiabatic deformation heating to remove the effect of excessive flow softening.

### 2.3. Observing Microstructure

Optical microscope (OM, MA 200, Nikon, Japan) is used to observe the microstructure of as-cast billet and deformed specimens by a compressive test. The microstructure observation by using OM is conducted after electro-etching samples with Barker's reagent (5 mL HBF4, 200 mL distilled water) under a condition of 25 V for 70 s at room temperature by using a Lectropol 5 machine (Struers, Kobenhavn, Denmark). In order to evaluate

not only macro scale microstructures, but also sub-micron scale microstructures, such as dislocations, stored energy, and restoration, the EBSD method is applied by using the Field emission scanning electron microscope (FE-SEM, QUANTA 200F, FEI, Hillsboro, OR, USA). A sufficiently large area (at least larger than 1 mm × 1 mm) was observed in those OM and EBSD analyses to secure the reliability of data. The obtained EBSD data was analyzed using TSL OIM software (EDAX-AMETEK, Berwyn, PA, USA). The FE-SEM sample preparation was performed by polishing samples with SiC papers (from #800 to #4000), and then electro-polished under the condition of 35 V for 30–60 s at −20 °C in 2% perchloric acid based on an ethanol solvent.

## 3. Results

### 3.1. Flow Stress-Strain Curves Behavior

Figure 1 shows flow stress-strain curves of Al-6Mg and Al-8Mg alloys at various strain rates and temperatures. As the temperature increases, and strain rate decreases, the flow stress decreases. When the two Al-Mg alloys are compared at the same working condition, as the Mg contents increases, the peak stress increases. However, in Figure 1, when the compressive test is conducted at the temperatures above 350 °C in the range of low strains, the flow stress drop occurs. Beyond the peak strain, the increase in stress is not shown, but the stress does not change or decrease. Moreover, in the Al-8Mg alloy (higher Mg content), this phenomenon becomes more evident than the Al-6Mg alloy.

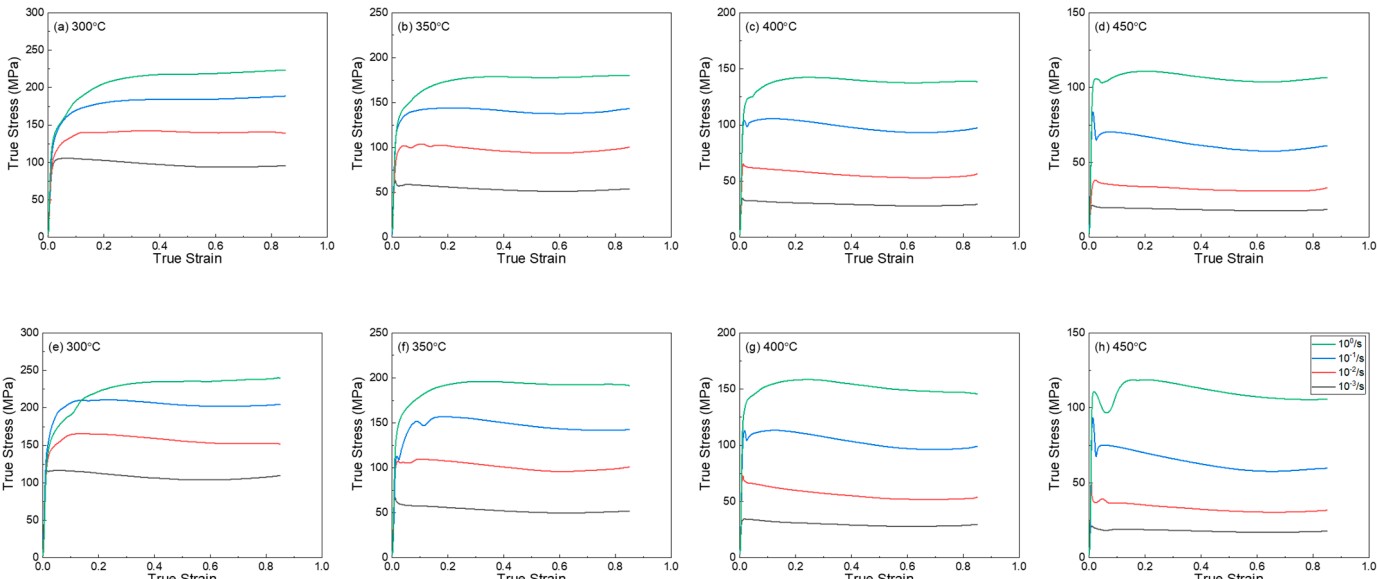

**Figure 1.** Flow curves of (**a**–**d**) Al-6Mg and (**e**–**h**) Al-8Mg obtained various temperatures and strain rates.

### 3.2. Processing Maps

The compressive test results are analyzed using the dynamic materials model suggested by Prasad [16]. This model proposes a power dissipation factor ($\eta$), which can help evaluate the energy conversion efficiency of deformed specimens. Under the hot deformation conditions, the input power to the specimens can be described as a sum of G and J. G is the term representing the energy conversion into heat and J is the dissipated energy contributing to microstructural change [17,18].

$$\mathrm{P} = \mathrm{G} + \mathrm{J} = \int_0^{\dot\varepsilon} \sigma \mathrm{d}\dot\varepsilon + \int_0^{\alpha} \dot\varepsilon \mathrm{d}\sigma \tag{1}$$

where $\sigma$ is the flow stress, $\dot\varepsilon$ is the strain rate, $\varepsilon$ is the strain, and $\alpha$ is an exponent. The

strain rate sensitivity can be adopted by m, which is derived from the relationship between G and J.

$$m = \frac{dJ}{dG} = \frac{\dot{\varepsilon}d\sigma}{\sigma} \tag{2}$$

G and J are derived and expressed by the stress, strain, strain rate, and m value.

$$G = \int_0^{\dot{\varepsilon}} \sigma d\dot{\varepsilon} = \frac{\sigma\dot{\varepsilon}}{1+m} \tag{3}$$

$$J = \int_0^{\sigma} \varepsilon d\sigma = \frac{m\sigma\dot{\varepsilon}}{1+m} \tag{4}$$

In the ideal condition of maximum power dissipation, m is 1, and the maximum value of J is half of P. Therefore, $\eta$, which is the power dissipation efficiency, is defined by Equation (5). This equation means the dissipated energy efficiency. In other words, since the $\eta$ value is high, the conducted power is well distributed, and the hot deformation is stable.

$$\eta = \frac{J}{J_{max}} = \frac{2m}{m+1} \tag{5}$$

Ziegler has researched the stability of material in hot deformation, and suggested the equation that expresses the stability region [19].

$$\frac{dJ}{d\dot{\varepsilon}} > \frac{J}{\dot{\varepsilon}} \ or \ \frac{d\ln J}{d\ln\dot{\varepsilon}} > 1 \tag{6}$$

Using Equations (1), (4) and (6), the Ziegler parameter ($\xi$), which indicates the stability of plasticity, is defined below.

$$\ln J = \ln\left(\frac{m}{m+1}\right) + \ln\sigma + \ln\dot{\varepsilon} \tag{7}$$

$$\frac{d\ln J}{d\ln\dot{\varepsilon}} = \frac{d\ln(m/m+1)}{d\ln\dot{\varepsilon}} + \frac{d\ln\sigma}{d\ln\dot{\varepsilon}} + 1 \tag{8}$$

$$\xi(\dot{\varepsilon}) = \frac{\partial\ln(m/m+1)}{\partial\ln\dot{\varepsilon}} + m > 0 \tag{9}$$

Figure 2 plots several processing maps of Al-6Mg and Al-8Mg of this study, which describe η and ξ distribution. In Figure 2a,b, both Al-6Mg and Al-8Mg show the highest power dissipation efficiency at 400 °C and $10^{-3}$/s. When focusing on the Mg content effect, the maximum η value appears higher in the Al-8Mg (Figure 2b) than the Al-6Mg (Figure 2a). Not only the maximum η value, the η values in Figure 2b are generally larger than those in Figure 2a under the same hot working conditions. In both Al-8Mg and Al-6Mg processing maps, the power dissipation efficiency typically improved by increasing the deformation temperature and by decreasing the strain rate. The area covered by the blue dashes represents the plastic instability region where ξ maintains the negative value, and the other non-dashed areas indicate where ξ is positive. The results in Figure 2 may indicate that the flow stability in the Al-Mg alloy can be improved by adding more Mg content in the Al alloys since the dashed area is smaller in Figure 2a,b.

### 3.3. Microstructure

Figure 3 is the polarized OM and EBSD observation results of the microstructure after homogenization of the two alloys (non-deformed). In both alloys, equiaxed grains are observed. The average grain size measured using EBSD and the values are 155.02 ± 49.34 μm for Al-6Mg and 125.75 ± 35.10 μm for Al-8Mg. Figure 4 shows the deformed microstructures of Al-6Mg (Figure 4a,c) and those of Al-8Mg (Figure 4b,d) at 300 °C, $10^0$/s, where the plastic instability occurred, and 400 °C and $10^{-3}$/s, where the η values are the highest in the

processing maps in Figure 2. In the hot working condition of 300 °C, $10^0$/s, in both alloys (Figure 4a,b), shear bands are developed as the form of narrow bands. Moreover, while only the deformation band is observed inside the grain in Al-6Mg, whereas, in Al-8Mg, not only deformation bands formation inside the grain, but also some recrystallizations at grain boundaries are found. Now, in Figure 4c,d, the microstructures where the highest η value is obtained in both alloys are given. It is observed that dynamic recrystallization is formed at 400 °C and $10^{-3}$/s, where the energy dissipation efficiency is high in the two Al-Mg alloys.

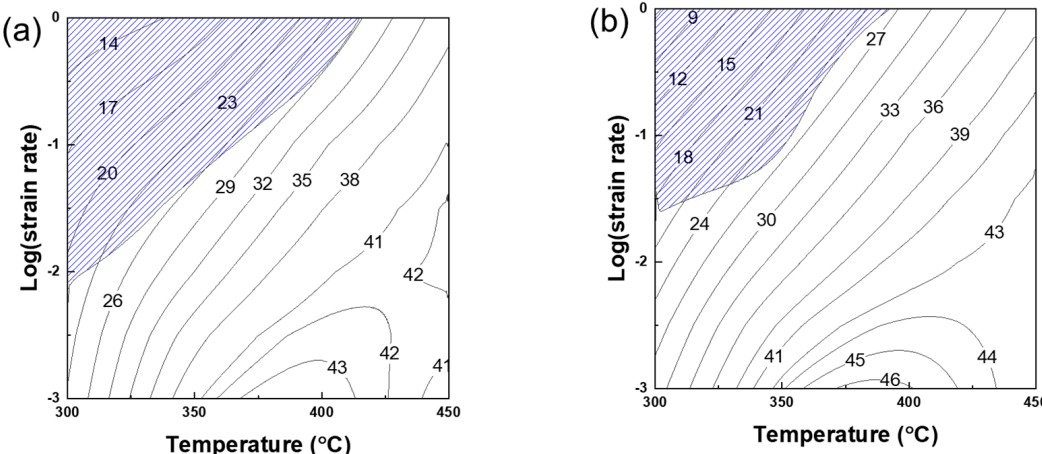

**Figure 2.** Processing map of (**a**) Al-6Mg and (**b**) Al-8Mg with 0.2 of a strain. The dashed area represents the instability region.

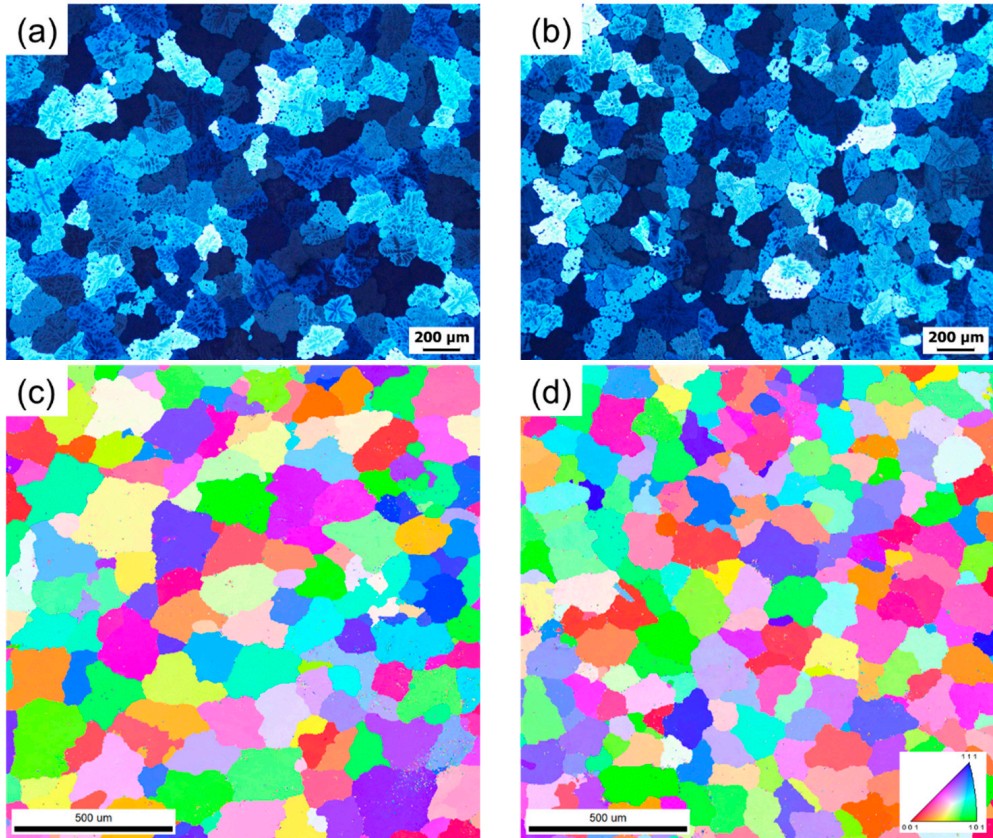

**Figure 3.** Grain structure of homogenized initial microstructure for (**a**) Al-6Mg, (**b**) Al-8Mg by an optical microscope, and (**c**)Al-6Mg, and (**d**)Al-8Mg by an EBSD IPF map.

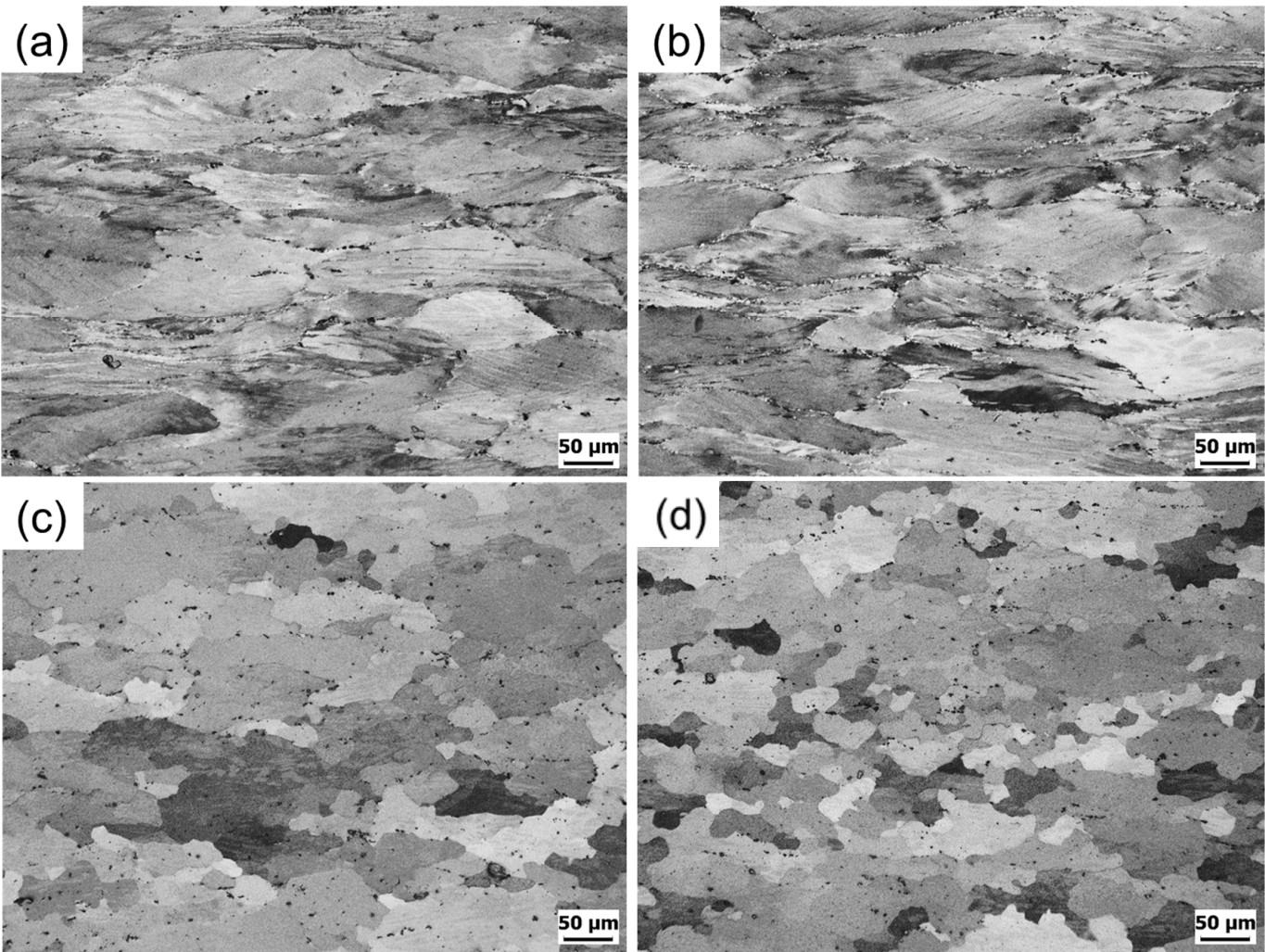

**Figure 4.** Optical micrographs of the as hot compressed microstructures in the (**a**) Al-6Mg and (**b**) Al-8Mg of 300 °C–$10^0$/s and (**c**) Al-6Mg and (**d**) Al-8Mg of 400 °C–$10^{-3}$/s at a strain of 0.8.

Figure 5 shows several kernel average misorientation (KAM) maps of Al-6Mg (Figure 5a,b) and Al-8Mg (Figure 5c,d). Similar to Figure 4, the KAM maps under the instability region are given in Figure 5a,c for Al-6Mg and Al-8Mg alloys, respectively. The KAM maps of maximum η conditions are suggested in Figure 5b for Al-6Mg and Figure 5d for Al-8Mg. In Figure 5a,c, it is clearly shown that the high local strain is present inside grains of both Al-6Mg and Al-8Mg alloys. This suggests that dynamic restoration rarely occurred under the instability condition in both alloys. When the two different Al-Mg alloys are compared, in this instability region, no remarkable difference is found in the KAM maps. The KAM maps under the highest η value conditions show the evidently smaller local strain (or absence of local strains) inside grains, as present in Figure 5b (Al-6Mg) and 5d (Al-8Mg). The dynamic recrystallization behavior observed in Figure 4c,d can provide one plausible explanation for this low local strain distribution in Figure 5b,d. When the two Al-Mg alloys are compared in the maximum power dissipation efficiency region, it appears that Al-8Mg may have some wider local strain area, where green local strains are widely observed. However, as the deformation temperature increases and the strain rate decreases, it is shown that grains of a relatively similar size are uniformly distributed, and it is clear that the gradient of the KAM value is relatively decreased. However, the higher the Mg content is, the less the decrease in the gradient of the KAM value is when observed by comparing Figure 5b,d. Remarkably, in Figure 5d, it is observed that a portion with a high KAM

value is concentrated inside a specific grain. However, such a phenomenon is not observed in Figure 5b.

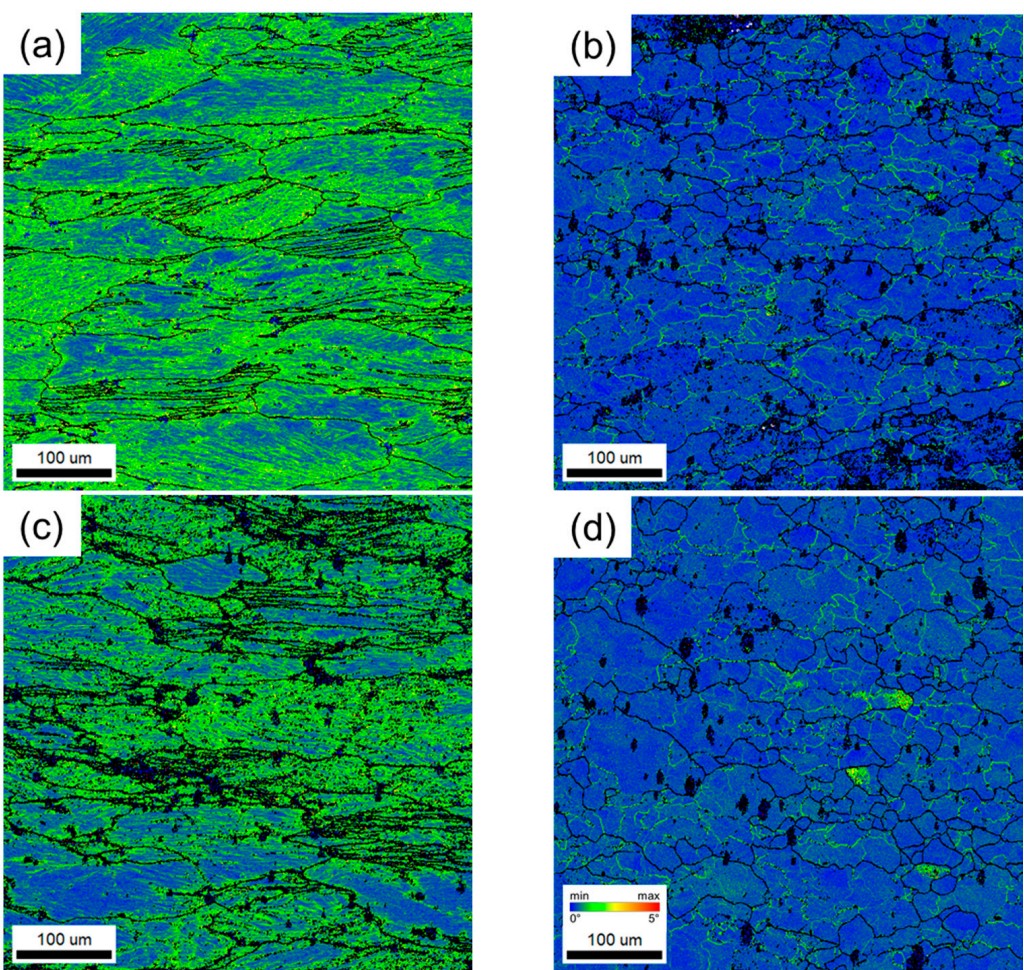

**Figure 5.** KAM maps of the (**a**) Al-6Mg, 300 °C, $10^0$/s, (**b**) Al-6Mg, 400 °C, $10^{-3}$/s, (**c**) Al-8Mg, 300 °C, $10^0$/s, (**d**) Al-8Mg, 400 °C, $10^{-3}$/s.

Figure 6 and Table 2 show the grain orientation spread (GOS) map and results for each compression condition. GOS is a value obtained by averaging the orientation difference between adjacent EBSD data points in one identical grain, so that it can provide local strain information like the case of KAM data in Figure 5. In other words, the GOS value represents the degree of average dispersion of the misorientation within one grain [20,21]. Therefore, some low GOS values can be obtained if some restoration, such as recrystallization and recovery, occurred in the chosen grain. In order to calculate GOS, it is required to determine individual grain boundaries, and, in this study, 15° is set to the misorientation to define a grain boundary. As the compression conditions changed from 300 °C, $10^0$/s to 400°C, $10^{-3}$/s, the average GOS value of Al-6Mg decreased from 1.724 to 0.582. Additionally, in the case of Al-8Mg, the average GOS value decreased from 1.631 to 1.08 in the two deformation conditions. Moreover, the fraction of low GOS region (GOS < 3°) increases from 0.031 to 0.231 in the Al-6Mg by changing from the instability region to the highest η region. In the case of Al-8Mg alloy, the fraction where GOS < 3° increases from 0.026 (instability region) to 0.278 (maximum η region).

## 4. Discussion

In Figure 1, generally, as the processing temperature increases, the stress decreases. This phenomenon is generally shown in material processing because the dislocation glide

becomes easier at higher temperatures and lower stresses [22–24]. The easy dislocation glide enables the energy required for deformation to be lower as the flow softening behavior occurs. After yielding, at a high temperature (upper than 350 °C), the softening of flow stress is shown, whether Mg contents are 6 or 8Mg, as present in Figures 4 and 6. Dynamic recrystallization may explain this flow stress drop in the flow curves in Figure 1. As Mg contents increase, this phenomenon becomes evident, which can be partially supported by the higher low GOS fraction area in Al-8Mg in Table 2. The Mg content in Al alloys seems to be deeply related to dynamic recrystallization (DRX) mechanisms. The increase in Mg contents in Al alloys not only leads to the strengthening of Al alloys by a solid-solution strengthening mechanism, but it appears to accelerate the DRX rate [25].

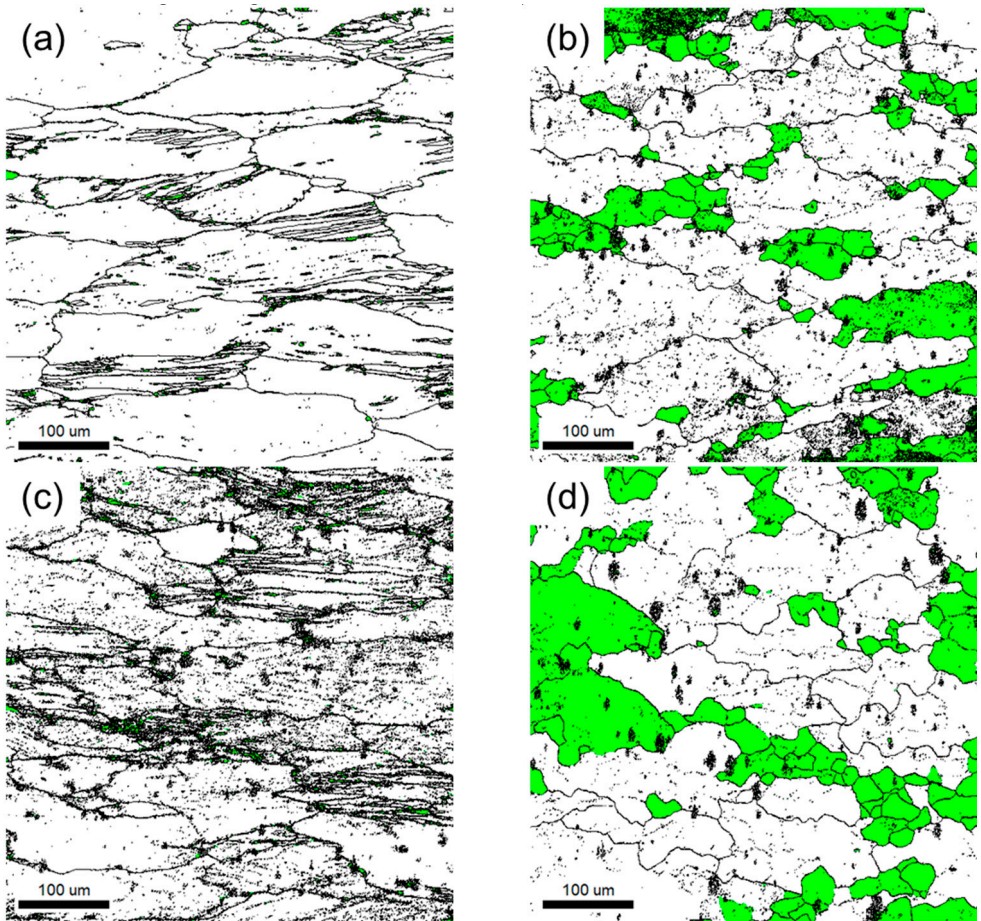

**Figure 6.** GOS maps of the (**a**) Al-6Mg, 300 °C, $10^0$/s, (**b**) Al-6Mg, 400 °C, $10^{-3}$/s, (**c**) Al-8Mg, 300 °C, $10^0$/s, and (**d**) Al-8Mg, 400 °C, $10^{-3}$/s. The green grains have average GOS under 3°.

**Table 2.** Grain orientation spread value for each compression condition.

| Process Conditions | GOS Value | GOS < 3° Fraction |
|---|---|---|
| (a) Al-6Mg, 300 °C, $10^0$/s | 1.724 | 0.031 |
| (b) Al-6Mg, 400 °C, $10^{-3}$/s | 0.582 | 0.231 |
| (c) Al-8Mg, 300 °C, $10^0$/s | 1.631 | 0.026 |
| (d) Al-8Mg, 400 °C, $10^{-3}$/s | 1.080 | 0.278 |

The effect of Mg content on recrystallization can also be observed on the processing map (Figure 2). The power dissipation efficiency is an indicator of whether the input energy is adequately converted to material deformation or microstructure evolution. In general, as DRX is activated, the power dissipation efficiency increases by consuming strain localization, or stored strain energies in the material [15,26,27]. Therefore, it is anticipated

that the efficiency generally increases by increasing hot working temperatures as dynamic restoration is expedited at elevated temperatures. In addition, there is a general tendency that reduction in the power dissipation efficiency is promoted by non-uniform deformation under faster strain rate conditions, as present in Figures 2 and 4 with the deformation band formations.

Furthermore, as shown in Figure 4, the amount of DRX is larger in the Al-8Mg alloy than the Al-6Mg alloy under the maximum η condition. This phenomenon may influence the change in energy dissipation efficiency and the development in the instability region in the processing map. The processing maps show the highest processing efficiency between 350–400 °C in the two alloys with the evolution of dynamic recrystallization.

The DRX behavior appears to affect the flow behavior of Al-Mg alloys at elevated temperatures. First, the DRX evolution was observed in the 400 °C deformed samples by using OM and EBSD (Figure 4c,d and Figure 6b,d). However, no evidence of DRX formation was shown in the compression tested samples at 300 °C (Figure 4a,b and Figure 6a,c). With coincident to these results, while the 300 °C flow curves in Figure 1 did not exhibit a peak stress and subsequent flow drop, the 400 °C flow curves showed some flow drop after the peak strain. Additionally, the degree of flow drop was larger in the Al-8Mg than in the Al-6Mg at temperatures above 400 °C. This suggests a considerable energy absorption that occurred in the 400 °C deformed samples to initiate DRX, whereas DRX was rarely activated in the 300 °C tested samples. The stored energy absorption during the compression tests at 400 °C is also clearly shown in the KAM maps in Figure 5.

The GOS maps support this argument as well (Figure 6 and Table 2). Specifically, the fraction of the low GOS value area is much larger in the 400 °C conditions when compared to the 300 °C conditions. Since the GOS value indicates the degree of local micro-strains inside each grain, it has been generally accepted that the low GOS value area (below 2° or 3°) represents the restored region by recovery and recrystallization. In other words, it may simply say that the low GOS value area indicates the DRX fraction of the deformed samples in Figure 6. Therefore, in Table 2, the DRX was formed slightly more in Al-8Mg than in Al-6Mg at 400 °C. However, when the 300 °C data of the two Al-Mg alloys are compared in Table 2, it is noted that both the average GOS value and the low GOS fraction were comparable between the two different Al-Mg alloys at 300 °C. This may suggest that the Mg content effect is not important for microstructure evolution in Al-Mg alloys when the deformation condition does not allow DRX to be activated. In the DRX activated region (400 °C deformation condition) in Table 2, the average GOS value was higher in the Al-8Mg alloy than the Al-6Mg alloy, which proposes that the generated local micro-strain remains more in the higher Mg-contained Al alloy after the hot deformation despite its larger DRX fraction. It is believed that dynamic recovery becomes substantial as the Mg content decrease in Al-Mg alloys, so that faster local strain reduction occurs in the low-Mg level Al alloy.

Hence, in hot deformation of Al-Mg alloys, the Mg content seems to have opposite effects on the dynamic restoration behaviors, namely dynamic recovery and dynamic recrystallization under hot working conditions where dynamic restoration mechanisms are activated. Since Mg solutes in the Al matrix hinder the movements of dislocations and let strain energies be accumulated easily, the degree of stored energy more rapidly increases by adding more Mg content in Al alloys [28]. This expedited energy storage leads to the faster DRX formation. However, since dislocation movements are disturbed by Mg atoms, the decrease of Mg content in the Al matrix appears to provoke a mutual interaction between dislocations, such as dislocation annihilation, so that dynamic recovery becomes dominant as the Mg content decreases.

## 5. Conclusions

In this study, the high-temperature compression behaviors of new Al-Mg alloys with high Mg content (Al-6Mg and Al-8Mg) were investigated through processing maps and a microstructure observation. These findings led to the following conclusions.

1.  The flow curves show that the maximum strength increased as the Mg content increased. In addition, a rapid softening behavior of flow stress was observed after the yield point at a temperature of 350 °C or higher. Dynamic recrystallization may explain this phenomenon.
2.  According to the results of the processing maps, it is observed that, as the Mg content increases, the maximum power dissipation efficiency increases, whereas the plastic instability region area decreases. The level of power dissipation efficiency increases with the activation of dynamic restoration, especially dynamic recrystallization (DRX). It appears that the inhomogeneous deformation is suppressed by the acceleration of DRX in the Mg content under the dynamic restoration-activated conditions. Deformation band formation is evident in the region where the plastic instability occurs with a low power dissipation efficiency.
3.  The kernel average misorientation map visually indicates that the remaining local strain is much smaller in the maximum power dissipation condition than the low condition in both Al-6Mg and Al-8Mg alloys. The accumulated local strains during hot deformation of the maximum efficiency region appears to lead to the DRX formation.
4.  In a grain misorientation spread (GOS) analysis, both the average GOS value and the low GOS value fraction (<3°) are used. While the Mg content does not affect the microstructure evolution when dynamic restorations are not activated, it seems that the higher Mg concentration in Al-Mg alloy boosts the DRX rate and retards dynamic recovery by inhibiting dislocation movements.
5.  Further research needs to be done by comparing the plastic deformation behavior and processing map results for each temperature and strain rate domain to reach high reliability in mass-scale products. This research might help design lightweight automotive components, such as aluminum forged parts (such as arm, rod, knuckle, axle, etc.).

**Author Contributions:** Conceptualization, N.-S.K. and B.-H.K.; methodology, N.-S.K.; software, N.-S.K., S.-Y.Y.; validation, K.-H.C., S.-Y.Y.; formal analysis, N.-S.K., S.-Y.Y., and B.-H.K.; investigation, N.-S.K.; resources, N.-S.K., Y.-O.Y., B.-H.K., and S.K.K.; data curation, N.-S.K., S.-Y.Y.; writing—original draft preparation, N.-S.K.; writing—review and editing, S.-H.H., Y.-O.Y., and H.-K.L.; visualization, N.-S.K.; supervision, S.K.K., S.-K.H.; project administration, B.-H.K.; funding acquisition, S.K.K. All authors have read and agreed to the published version of the manuscript.

**Funding:** The APC was funded by Korea Institute of Industrial Technology, Internal Budget.

**Data Availability Statement:** Data available in a publicly accessible repository.

**Conflicts of Interest:** The authors declare no conflict of interest.

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
