# Peer review of "Hot Compression Behavior of New Al-6Mg and Al-8Mg Alloy with Improved Hot Workability Fabricated by Direct Chill Casting Method"

_metals, doi:10.3390/met11020288_

Round 1
Reviewer 1 Report
Dear Authors,
you work seems to be valuable, although some improvements must be made:
- The language should be corrected, mainly the style but also the grammar.
- There is no statistical analysis which could support obtained results and conclusions. Were the tests repeated?
- Subsections in Section 2 should be named as 2.1., 2.2 etc.
- In the Discussion section, references to figures and tables should be given.
- Conclusions should be given more precisely, with the better explanation of obtained results.
- What are the indications for the industry?
- Further research should be also indicated in the conclusions.
- There are only five references from last five years. Please include more references published in recent years.
Author Response
Dear reviewer.
I am pleased to submit our revised research article entitled “Hot compression behavior of new Al-6Mg & Al-8Mg alloy with improved hot workability fabricated by direct chill casting method” by Nam-Seok Kim, Kweon-Hoon Choi, Seung-Yoon Yang, Seong-Ho Ha, Young-Ok Yoon, Bong-Hwan Kim, Hyun-Kyu Lim, Shae. K Kim, Soong-Keun Hyun for consideration for publication in Metals special issue.
We greatly appreciate the comments and suggestions made by the reviewers. These are reasonable and constructive. Accordingly, our paper has been substantially revised.
This manuscript is the revised version from the original manuscript (submitted on Jan.15 2021). All revisions in the manuscript were made based on suggestions and comments from three reviewers of the original manuscript. All changes to the manuscript can be easily tracked through the track change function. Additionally, specific answers to those suggestions and comments from the three reviewers are given in the “Detailed response to the reviewer’s comments”.
See attachment for revised manuscript.
- The language should be corrected, mainly the style but also the grammar.
[Answer] The revision version was corrected by a colleague obtained a doctorate degree in Metallurgical Engineering from the United States. - There is no statistical analysis which could support obtained results and conclusions. Were the tests repeated?
[Answer] This manuscript was written after repeat testing by using a fully homogenized sample. We have added that to our experimental method. (line 83-84, track change off) - Subsections in Section 2 should be named as 2.1., 2.2 etc.
[Answer] This is a simple marking error. I changed 1, 2, 3 to 2.1, 2.2, 2.3. - In the Discussion section, references to figures and tables should be given.
[Answer] We added reference to the discussion section and added figure 7. This figure explains the relationship between the change in magnesium content and dynamic recrystallization in aluminum alloys. - Conclusions should be given more precisely, with the better explanation of obtained results.
[Answer] The conclusion has been revised. We tried to more accurately express the conclusions obtained through this experiment. - What are the indications for the industry?
[Answer] The subject of the special issue is related to light-metal forming technologies for automobiles and aircraft. This paper is related to the hot deformation mechanism of high strength aluminum alloys, which needs to be understood well for forming technology. However, to convey that exact meaning, the word was deleted and the sentence was corrected to a more appropriate expression. (line 39-41) - Further research should be also indicated in the conclusions.
[Answer] We added further research in conclusion 5. - There are only five references from last five years. Please include more references published in recent years.
[Answer] I have added/modified the list of references to include the latest paper.

Reviewer 2 Report
Presented paper is important study on aluminum - magnesium alloys. This is a good paper with interesting study. All topics are well presented and discussed. My only concern is title of this paper. Reviewer suggests to change of alloys indication, because Al-6 & 8Mg is misleading.
Author Response
Dear reviewer.
I am pleased to submit our revised research article entitled “Hot compression behavior of new Al-6Mg & Al-8Mg alloy with improved hot workability fabricated by direct chill casting method” by Nam-Seok Kim, Kweon-Hoon Choi, Seung-Yoon Yang, Seong-Ho Ha, Young-Ok Yoon, Bong-Hwan Kim, Hyun-Kyu Lim, Shae. K Kim, Soong-Keun Hyun for consideration for publication in Metals special issue.
We greatly appreciate the comments and suggestions made by the reviewers. These are reasonable and constructive. Accordingly, our paper has been substantially revised.
This manuscript is the revised version from the original manuscript (submitted on Jan.15 2021). All revisions in the manuscript were made based on suggestions and comments from three reviewers of the original manuscript. All changes to the manuscript can be easily tracked through the track change function. Additionally, specific answers to those suggestions and comments from the three reviewers are given in the “Detailed response to the reviewer’s comments”.
See attachment for revised manuscript.
-
Presented paper is important study on aluminum - magnesium alloys. This is a good paper with interesting study. All topics are well presented and discussed. My only concern is title of this paper. Reviewer suggests to change of alloys indication, because Al-6 & 8Mg is misleading.
[Answer] I have changed the title of our paper like below.
- Before correction
Hot compression behavior of new Al-6 & 8Mg alloy with improved hot workability fabricated by direct chill casting method - After Correction
Hot compression behavior of new Al-6Mg & Al-8Mg alloy with improved hot workability fabricated by direct chill casting method
- Before correction

Reviewer 3 Report
The english should be checked by a native English speaker. Sometimes the text is difficult to follow (lines136-146) or the meaning is not clear due to weird use of words (lines 227-244; what should mean a word "acquired" in this context?).
Lines 70-80: a description of the compression tests should be more thorough- What method was applied for strain measurement? What was the number of the replicated tests?
Line 99: it is Fig. 1, not Fig. 2.
Figure 1: please, add a legend on strain rates.
Figure 3: do not split the figure over two pages.
Figure 4 and 5: poor figures in blue color. Make them more clear.
A section on discution is just an additional description of the results, some of it was already done in section 3. What is the practical meaning of the presented results for the users? Please, give an explanation.
Author Response
Dear reviewer.
I am pleased to submit our revised research article entitled “Hot compression behavior of new Al-6Mg & Al-8Mg alloy with improved hot workability fabricated by direct chill casting method” by Nam-Seok Kim, Kweon-Hoon Choi, Seung-Yoon Yang, Seong-Ho Ha, Young-Ok Yoon, Bong-Hwan Kim, Hyun-Kyu Lim, Shae. K Kim, Soong-Keun Hyun for consideration for publication in Metals special issue.
We greatly appreciate the comments and suggestions made by the reviewers. These are reasonable and constructive. Accordingly, our paper has been substantially revised.
This manuscript is the revised version from the original manuscript (submitted on Jan.15 2021). All revisions in the manuscript were made based on suggestions and comments from three reviewers of the original manuscript. All changes to the manuscript can be easily tracked through the track change function. Additionally, specific answers to those suggestions and comments from the three reviewers are given in the “Detailed response to the reviewer’s comments”.
See attachment for revised manuscript.
- The english should be checked by a native English speaker. Sometimes the text is difficult to follow (lines136-146) or the meaning is not clear due to weird use of words (lines 227-244; what should mean a word "acquired" in this context?).
[Answer] The English expressions in this paper have been entirely corrected by colleagues. The line 136-146 has been revised and written to lines 142-154 and The line 227-244 has been revised and written to lines 255-265. (Line numbers are based on when the track change function is turned off.) - Lines 70-80: a description of the compression tests should be more thorough- What method was applied for strain measurement? What was the number of the replicated tests?
[Answer] We added the strain measurement method and the reliability of the experiment to the section 2.2.
- (line 77-79) Strain was used by calculating the engineering strain obtained by measuring between anvils as the true strain.
- (line 83-84) Additionally, the reproducibility of data was checked by repeating the same experiments at least three times. - Line 99: it is Fig. 1, not Fig. 2.
[Answer] This is a simple marking error, so I revised it in the manuscript. - Figure 1: please, add a legend on strain rates.
[Answer] In Figure 1. (a), I completed to add the legend. - Figure 3: do not split the figure over two pages.
[Answer] I modify the contents and figure not to split over two pages. - Figure 4 and 5: poor figures in blue color. Make them more clear.
[Answer] I modify the color, brightness, and contrast of images to make a clear figure. - A section on discussion is just an additional description of the results, some of it was already done in section 3. What is the practical meaning of the presented results for the users? Please, give an explanation.
[Answer] We revised the conclusion including the revised comments and English correction.

Round 2
Reviewer 1 Report
Dear Authors,
I am satisfied with the answers. Some corrections in text editing is necessary.
Author Response
Thanks for the kind review.
We checked the text editing again and upload an revised manuscript.
Reviewer 3 Report
Conclusions, point 5: The authors make the following statement:"Further research needs to be done by comparing the plastic deformation behavior and processing map results for each temperature and strain rate domain to reach high reliability. This research might help to design the light-weight automobile in the future." To reach high realibility on what issue? Which light-weight automotive components can be designed on the basisi of this data? Safety components? Fatigue-loaded components? Auxiliary structural members?
Author Response
Thanks for the kind review.
In mass-scale production, local Inhomogeneity may occur due to the difference in thickness of each part of the product during the plastic deformation process. We hope that this study can help reduce these cases.
So we have revised the sentence to clarify the meaning.
- Before correction
Further research needs to be done by comparing the plastic deformation behavior and processing map results for each temperature and strain rate domain to reach high reliability. This research might help to design the light-weight automobile in the future. - After correction
Further research needs to be done by comparing the plastic deformation behavior and processing map results for each temperature and strain rate domain to reach high reliability in mass-scale products. This research might help to design lightweight automotive components such as aluminum forged parts (such as arm, rod, knuckle, axle, etc.).